# Gender-Related Challenges in Educational Interventions with Syrian Refugee Parents of Trauma-Affected Children in Turkey

**DOI:** 10.3390/children6100110

**Published:** 2019-10-07

**Authors:** Melissa Diamond, Charles Oberg

**Affiliations:** 1A Global Voice for Autism, Minneapolis, MN 55442, USA; melissa@aglobalvoiceforautism.org; 2Division of Global Pediatrics, University of Minnesota, MN 55415, USA

**Keywords:** refugee, gender, trauma, educational intervention

## Abstract

Since 2012, more than three million Syrian refugees have fled to Turkey. While these refugees vary in socioeconomic background, it is notable that 50% of Syrian refugee children in Turkey display symptoms of post-traumatic stress and that more than 663,138 of these children between the ages of six and seventeen are not enrolled in school. For those children who are in school, high levels of trauma have significant implications for the education system as trauma alters the brain and affects the way children learn. A Global Voice for Autism is an international non-governmental relief and development organization that exists to equip teachers and families in conflict-affected communities. Its intent is to support the development and success of children with autism and trauma-related behavioral challenges in their classrooms, home, and communities. The instabilities inherent in the Syrian refugee experience pose a number of challenges to the organization’s effective implementation of programming. The experiences of refugees in Turkey are highly gendered. Therefore, a qualitative gender analysis was conducted to address and better understand the challenges faced when carrying out these educational interventions. The article examines domestic violence, sexual violence, and masculinity as gender-driven constructs that influence how refugees experience trauma. In addition, structural issues in existing support systems all present significant challenges to Syrian refugee parents that impede effective program implementation. It is imperative to assess structural issues in existing support services to address these challenges and to successfully carry out meaningful and impactful programming. This Brief Report provides a series of recommendations in order to ameliorate these challenges and increase the efficacy of educational interventions with Syrian refugee parents of trauma-affected and vulnerable children in Turkey. It concludes with a call for policy changes that protect refugees from deportation when accessing support services and a network of services that do not require residency permits. It calls for increased integration of parent trauma support in educational intervention trainings and the creation of safe spaces where mothers and fathers can discuss their own trauma and challenges in the hope of significantly enhancing program efficacy.

## 1. Introduction

Since 2012, more than three million Syrian refugees have fled to Turkey. While these refugees vary in socioeconomic background, it is notable that 50% of Syrian refugee children in Turkey display symptoms of post-traumatic stress [1]. A significant number of these children are not in school and for those children who are in school, high levels of trauma have significant implications for the education system as trauma alters the brain and affects the way children learn.

Syrian refugees in Turkey have tended to settle in concentrated communities. This has some advantages for Syrians who, for the most part, do not speak Turkish and whose cultural differences pose challenges, particularly within the education system. Across Turkey, an estimated 663,138 refugee children between the ages of six and seventeen are not enrolled in school [2]. Until recently, many Syrian refugees hesitated to send their children to school due to a perception that their stay in Turkey was temporary. However, as families started to realize the permanency of their residence in Turkey, school enrollment has steadily increased.

Research shows that effective education actively contributes to children’s development of skills for coping and resilience, which are key skills in a population where research shows children experience post-traumatic stress. However, even when Syrian refugees are enrolled in school, their ability to access an inclusive and effective education is limited. Overcrowding in the schools and a lack of trained teachers pose challenges to the education system. Furthermore, unsupportive or unstable home environments can interfere with the quality of a child’s education [3]. These challenges have a particular impact on trauma-affected children, who learn differently due to brain changes triggered by trauma, and children with learning differences such as Autism Spectrum Disorder [4]. The effects of trauma on learning are influenced by the regulation of hormones, particularly cortisol and neurotransmitters in response to stress leading to a state of chronic persistent arousal. This triggers a “fight or flight” reflex that interferes with learning and the acquisition of new knowledge. This helps to explain how stress within the context of the learning experience, regardless of the cause of that stress, can result in alteration in attention, memory and other learning processes [5,6].

## 2. Gender and the Nature of Trauma

Refugees by definition are persons who have fled their country of origin for reasons of feared persecution, war, conflict, or generalized violence and, as a result, require protection [7]. Many have suffered trauma, with a residual impact on their health and well-being [8]. By virtue of the ways that men and women have different refugee experiences, displaced men and women also cope with trauma in different ways. Trauma complicates and interferes with daily life regardless of gender. However, the ways in which it complicates and influences supports that are necessary differ for men and women due to perceptions about the gendered needs of refugee populations. This can have dangerous implications.

For mothers, who are traditionally the primary caregivers within the Syrian population, an inability to adequately provide for their children during displacement can be traumatizing [9]. This trauma and inability to cope can result in decreased self-esteem and increased parenting stress, which inhibit mothers’ abilities to implement learned educational practices at home with confidence as well as their abilities to create safe, supportive home environments for trauma-affected children. In addition, a mother may be stigmatized for having a child with a disability and/or who has experienced significant trauma. In addition, typical home-based care roles prevent the formation of social groups in Turkey [10]. This isolation can exacerbate the manifestations of parental trauma in the home, while relegation to the home leaves mothers without access to psychological support.

Experiences in the home are one of the primary contributors to traumatic stress in children [11]. Insufficient parental trauma support can destabilize a child’s home environment through neglect and violence [12]. Even young children who were born in Turkey and have never experienced war in Syria are experiencing trauma as a result of the manifestations of parental trauma in their homes. Parents must have access to the support they need in order to create safe, healthy environments for their children, and such environments are crucial for the effective education of trauma-affected children.

### 2.1. Sexual Violence

Sexual violence has historically been used as a tool in conflict, and the Syrian conflict is no exception [13]. The use of rape as punishment in Syrian prisons has been documented, as has widespread sexual assault in refugee camps and sexual violence in the home [14]. In 2013 alone, the UN found 38,000 victims of sexual violence in Syria, including women, men, and children, despite significant underreporting.

Due to the stigma of sexual assault, emphasis on virginity, and the reality of honor killings, women hesitate to report experiences of sexual violence. For men, especially men who were once active in the Syrian conflict, sexual abuse is also a reality. The stigma of sexual violence is significant for both men and women in the community. For women, discovered incidences of sexual assault from someone other than a marital partner can lead to divorce, which is particularly threatening in a refugee context where there are fewer chances for a woman to achieve financial sustainability and support her children on her own [15]. Such discoveries may even result in honor killings [16]. For men, discovery of sexual assault can lead to community ostracism, blackmail, and loss of employment [17]. Even when not expressed or reported, these experiences can severely inhibit a caregiver’s ability to offer support to the children they are teaching or raising.

Women may also experience sexual violence from their husbands. This violence may be a continuation of ongoing sexual violence from before the conflict but may also materialize as a result of men’s unaddressed trauma and marginalization of masculinity in the refugee context [18]. While all sexual violence can result in trauma, sexual violence within the marital relationship may have the greatest impact on educational interventions for trauma-affected children due to the effects of ongoing sexual violence. Teachers who experience ongoing sexual violence may have less ability to concentrate on the needs of their students in the classroom, while sexual violence between parents at home alters family dynamics and can create a dangerous living environment for children. Finally, for refugee families who have lost everything, the pressure to earn money can force children into early marriage, taking them out of school and making them less likely to ever return [19].

### 2.2. Domestic Violence

Domestic violence occurs in a variety of ways for Syrian refugees living in Turkey. In some families, domestic violence is a continuation or a worsening of domestic violence perpetrated in Syria, while for others it is a new phenomenon. Both men and women perpetrate this violence, and children are often the victims of parental rage. Men are typically thought of as the perpetrators of domestic violence. However, women, lacking mental health support and grappling with the stresses of insecurity and isolation, often take out their frustrations on their children [20]. This creates a sense of insecurity for children at home, inhibits the efficacy of positive reinforcement approaches, and can exacerbate the trauma experienced by children [21]. Additionally, many Syrian women who flee to Turkey without husbands marry Turkish men, who often take them as second wives [22]. These situations leave women and their children vulnerable to abuse, and they are often tasked with household maintenance. Because Turkey does not recognize polygamous marriages, women in these marriages are especially vulnerable to abuse. Many Syrian women enter into these arrangements as a means of survival and cannot leave their husbands due to financial dependence [22].

One mother in A Global Voice for Autism’s program reported that, when she told her parents that her husband beat her so severely that she required hospitalization, her parents had asked her what she had done to upset her husband and told her to be more obedient in the future [23]. Without safe home environments that meet their own needs, parents cannot concentrate on the needs of their children. It is notable that in all of these scenarios, there is a trend of victim blaming by the broader community, which leads to a hesitancy in reporting or seeking support [24].

Given the strong gender norms in Syrian culture, aid agencies often dismiss sexual abuse and domestic violence as cultural factors that are not within the parameters of their mandates to support displacement-related needs [25]. This view not only allows domestic violence to continue without intervention, but also results in detrimental impacts on numerous other factors, such as children’s education, that aid organizations seek to address in times of conflict.

### 2.3. Masculinity

Historically, displacement has presented opportunities for gender roles to be renegotiated within displaced communities. Because of this, many community organizations in Turkey seek to challenge established gender roles within Syrian culture that define men as breadwinners and women as caregivers. In the camps, women are assigned leadership roles to challenge traditional gender structures. Outside of the camps, women’s employment schemes aim to shift breadwinner dynamics within the household [26].

However, when focused solely on women, such interventions have detrimental effects on men that can spill over into negative consequences for wives and children. Without access to support and opportunities, men’s pent-up frustrations can transform into domestic violence in an attempt to regain control amid feelings of marginalization [27]. Even without contrived shifts in gender roles, displacement challenges Syrian men’s conceptions of their own masculinity. Relocation to Turkey coincides with a loss of meaningful work, demotion to refugee status, and limited opportunities to provide for a family and make personal decisions [28].

These losses are coupled with increased time spent consuming media. Unemployed and seeking connections to home, refugee men access more televised media than ever before. Much of the media consumed is related to the war and emphasizes militaristic and violent conceptions of masculinity in its portrayals of fighting. However, even international news coverage of refugees reinforces these concepts of masculinity [29]. In depicting women and children as vulnerable and innocent and men as violent and dangerous migrants, the media reinforces understandings of masculinity that link it to violence [30].

Without productive outlets such as employment, and no support in the development of new masculine identities, masculinity can manifest as violence [31]. However, even when this does not occur, traditional narratives about Syrian masculinity can discourage fathers from engaging in caregiving activities. This makes it more difficult to include fathers in educational interventions. Because the practices that A Global Voice for Autism uses emphasize consistency, fathers’ lack of engagement can lead them to act in ways that counteract efforts to support their children’s learning using evidence-based practices [32].

### 2.4. Qualitative Gender Analysis

In order to understand and address the challenges that A Global Voice for Autism faces when carrying out these educational interventions, a qualitative gender analysis was conducted. Utilizing a combination of primary and secondary data, essential findings were identified related to the aforementioned barriers and their implications. Primary data was collated from A Global Voice for Autism’s past programs for Syrian refugees in Turkey along with data and information from secondary sources from partner organizations, researchers, and other community actors. Data collection methods included structured and unstructured narrative interviews, surveys, indices, video analyses, incident reports, and structured observation sessions. Of those families participating in educational support programing, 23% of families experienced past or present food and shelter insecurity, 27% reported experiencing domestic violence, and 14% reported experiencing sexual violence [33].

## 3. Structural Issues in Existing Support Systems

Children have specific rights, as articulated in the UN Convention on the Rights of the Child [34]. It has been thirty years since its adoption by the UN General Assembly, and it is as necessary now as when it went into effect. The convention articulates the principles of protection from harm, provision of basic needs, and the active participation of children as rights holders. Specifically, Article 22 states that countries must take appropriate measures to protect and assist all families of refugee children. It is from this context that structural deficiencies in existing support systems are explored [35].

Structural issues within Turkey exacerbate the impact of trauma and domestic and sexual violence in the Syrian refugee population. Although Turkey’s public hospitals offer free healthcare to residents, language barriers limit the ability of Syrian residents to access psychological support. Additionally, many public services that are theoretically available require that users hold a residency permit for their area of residence. However, the exorbitant costs of these residency permits are cost-prohibitive to most Syrian refugees, and an estimated 81% of women living outside of the camps do not hold permits [36]. Because of this, Syrians risk deportation to camps or back to Syria if they access the services that are theoretically available to address domestic and sexual violence. This is a particular vulnerability for women and children for whom the lack of privacy and threatened bodily integrity in the camp cause more immediate issues than they do for men.

In addition to the challenges that exist within the public support system, the private organizations that provide such supports to Syrians are often riddled with corruption. Family name, conflict party, and bribery all play roles in determining who has access to these services [37]. Even when people are in a social position to access these services, confidentiality is often compromised. Hence these services are not accessed due to the breech of confidence in the eyes of Syrians suffering from stigmatized experiences.

## 4. Components of Effective Educational Interventions

In order to address the aforementioned challenges within educational settings for Syrian refugees, A Global Voice for Autism works to equip communities with the skills and support they need to support trauma-affected children and children with autism in their classrooms, homes, and communities. Recognizing the interplay between home environment and effective education, the organization takes a holistic approach to their mission. Parents and teachers are trained in evidence-based practices for inclusive education and autism support. Parents and teachers then work together in cooperative groups to implement learned practices and also implement these practices at home and in their classrooms. Programming for Syrian refugee families in Turkey includes teacher trainings in inclusive education. Parent trainings aim to foster healthy home environments for affected children. In addition, sessions are designed to facilitate family and teacher support, self-development groups, and community education to increase understanding of trauma and autism. The programmatic components are standardized but are tailored to the composition of the group and the target setting, be it in the classroom, home, or community.

### 4.1. Parent & Teacher Training Cooperatives

Parents and teachers of primary-school-aged children are trained in globally recognized evidence-based practices for supporting their children with autism at home and in the community. Particular emphasis is placed on developing and improving independent methods of communication, reducing challenging behaviors, and teaching new skills.

### 4.2. Parent, Teacher, and Sibling Support and Self-Development Groups

These groups provide participants with safe spaces to share and reflect upon their experiences of supporting trauma-affected loved ones and loved ones with autism while gaining a deeper understanding of the experiences of those affected. Participants increase their self-confidence, improve their self-care, and discover ways to integrate their strengths and interests into their daily lives.

### 4.3. Community Education

These sessions educate community members about autism and trauma in order to reduce community stigma and increase understanding. They are offered to doctors, teachers, government officials, and general community members in order to decrease the marginalization of these individuals within these communities.

## 5. Impact of Gendered Challenges on Interventions

The instabilities inherent in refugee life pose a number of challenges for the organization’s effective implementation of programming. The gender challenges identified significantly impact A Global Voice for Autism’s ability to carry out educational interventions for trauma-affected children in the following ways:

### 5.1. Attendance and Participation

Attrition rates for participating families and teachers in A Global Voice for Autism’s programs in Turkey are high. The most frequently reported reasons for attrition are “required by spouse” and “inability to access program.” In the attrition cases in the past Turkey programs, 78% occurred in families where a family member had reported issues of domestic violence. Incidents at home, which often interlink challenges to masculinity with domestic and/or sexual violence, are primary reasons for attrition and, therefore, inhibitors to the success of the program [38]. Furthermore, these experiences and challenges pre-occupy families and take precedence over program participation. In order to attend and actively participate in the program, families must have their primary needs met [39].

### 5.2. Counterproductive Modeling and Behavioral Strategies

Modeling is a common behavior for children. When children see unhealthy and violent coping mechanisms at home, they may practice these behaviors. This is a particular issue when it comes to the interventions that are used, many of which rely on teaching through modeling skills and reinforcement of modeling behavior. Furthermore, consistency is key to the efficacy of these strategies, and when violence or parental disengagement occurs at home, it limits the efficacy of the home-based interventions in support of the education of trauma-affected children [40].

### 5.3. Trauma Replication

Existing trauma for trauma-affected children is perpetuated and exacerbated by instability in the home. As one of the strongest predictors of child performance, adverse childhood experiences (ACEs), such as domestic violence, parental disengagement, and displacement affect a child’s ability to learn new skills and engage with their environments [40,41]. Outcomes of these ACEs can include lack of trust. In order for educational interventions for trauma-affected children to be effective, trust of parent and teacher caregivers is essential [42].

## 6. Recommendations

Based on the aforementioned gendered challenges to educational interventions in the Syrian refugee population in Turkey, the following are recommended:Refugee support organizations should advocate for a policy within Turkey that provides refugees seeking emergency support services, such as psychological support, legal support, and domestic and sexual violence support, with status immunity. This means that no refugee or “guest” will risk deportation or punitive measures when accessing these support services.In concurrence with the above policy recommendation, there should be the establishment of a network of existing organizations that agree to support all refugees in need, regardless of their legal status, so that parents and teachers can access the services they need in order to achieve a level of wellbeing that enables them to effectively support children with autism and trauma-affected children.A Global Voice for Autism, and other organizations supporting educational interventions in Syrian refugee populations in Turkey, should increasingly integrate their support for affected children with parental support programs that teach parents to address their own trauma so that they can focus on addressing the needs of their children. Such programs will be self-reinforcing because, as parents reach a level where they are able to implement effective practices in support of their children, parenting will become easier and parenting stress will decrease.In order to address the isolation experienced by many refugee mothers, as well as the lack of safe spaces for fathers to discuss trauma, partnerships should be established with other community organizations to expand the availability of women’s and men’s groups for parents of trauma-affected children to create spaces for connection and experience processing with others who have similar experiences.

## 7. Conclusions

The creation of stable, secure environments for children is essential to the success of educational interventions for trauma-affected refugee children. Trauma alters the brain in ways that result in learning differences, and a sense of security is integral to effective trauma-informed education and care. In the Syrian refugee population in Turkey, creating these secure environments for learning is a challenge due to gender-related issues of masculinity, domestic violence, and sexual violence. This, in conjunction with the persistent structural issues in existing support systems, inhibit access to appropriate support for the population. In order to address these issues, and to foster safe environments where stakeholders can support trauma-affected children in their education, policies in Turkey must ensure the safety of refugees who access support services. Organizations must also maintain refugee confidentiality regardless of legal status, and A Global Voice for Autism and other community organizations must increasingly integrate parent trauma support into child trauma support trainings while providing access to networks of support for both men and women caregivers in the community.

To achieve such results, the availability of funding and resources must be discussed. In 2016, the European Union (EU) signed an agreement with Turkey to assume responsibility for migrants who initially entered Europe through Turkey. It was intended to limit the influx of families entering the EU through Turkey. The EU allocated three billion Euros in aid to Turkey to assist Syrian migrant communities. However, there is concern that the majority of funds have been allocated to the administrative costs of international civil society organizations rather than to direct aid for the refugees [43]. A comprehensive review should be undertaken to ascertain how the funding has been allocated to date, with the development of a strategy to promote program collaboration outlined above to enhance the direct care of Syrian refugee children.

Finally, though the difficulties faced by refugee children are immense, there is a growing international awareness that action needs to be taken to address displacement related challenges. In 2016, the UN General Assembly adopted the New York Declaration for Refugees and Migrants [44]. In 2018, the Global Compact for Safe, Orderly and Regular Migration and the Compact on Refugees were finalized [45]. Together they hopefully provide a global framework to support international efforts to address the crisis. It will be imperative that the United Nations focuses on the efforts in Turkey so as to operationalize the Compacts and optimize the care of Syrian children in Turkey.

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
