# Peer review of "Gender-Related Challenges in Educational Interventions with Syrian Refugee Parents of Trauma-Affected Children in Turkey"

_children, 2019, doi:10.3390/children6100110_

Round 1

Reviewer 1 Report

The authors make a sound contribution to the knowledge about barriers encountered by trauma-affected Syrian Refugee children and vulnerable children in Turkey concerning educational interventions, uniquely identifying challenges concerning gendered experiences of their parents. The authors make a strong case for tackling the post-traumatic stress experienced by children, both in and not attending school. They outline diverse forms of violence and issues concerning masculinity and changing gender roles within the refugee experience. Importantly, the analysis challenges aid agencies not to accept sexual abuse and violence as gender norms of Syrian culture (lines125-129), but instead to widen their parameters accordingly. Identifying structural issues in existing support systems, particularly the lack of resources available to refugees to obtain residency permits, is a particular strength. The recommendations are appropriate and their conclusions valuable.

Clearly, in a Brief Report, not all angles of the study can be outlined. Nevertheless, there are some areas which, if elaborated, would enhance the report’s significance and add weight to the recommendations.

Ethical issues

Although the report reflects on the impact of parental conflict upon children, the children are presented as passive, rather than as rights-holders whose views must be taken into account, as per the UN Convention on the Rights of the Child. Understanding their coping mechanism and resilience, opinions and wishes, would throw additional light on the intra-family dynamics. Identifying the views of boys as well as girls, and those of disabled and especially vulnerable children, would be important.

Contextualisation

The study’s contextualisation could be elaborated, in particular regarding the diverse issues identified in the refugee camps and in areas where Syrian refugees have gathered to live in close-knit communities. Cultural differences in the education systems (lines41,42) are not elaborated. The different vulnerabilities and opportunities of each should be clarified, not least by distinguishing asylum seekers and recognised refugees, who have a different status and rights. Reference is also made to customs in the Turkish communities in which Syrian refuges live, and it would be important to highlight clearly which issues only affect the refugees, e.g. being so desperate as to become the illegal second wife of a Turkish man. Would this also apply to poverty-stricken Turkish women and girls?

Educational interventions

The components of such interventions are referred to but they are not described in any depth (lines175ff). Are they different in camps and in the community? The paper would benefit from reporting how the potential stigmatisation of children (and adults) receiving specialist educational support is being avoided. It would be worthwhile to provide a reference to any evaluation of such interventions.

The study

The paper states that a gender study has been carried out (line20) but little detail of the study is provided. Timing, scope, methodology, tools, analysis, selection of participants/control group should be clearly stated. If such a study was in fact not carried out, a description of a more qualitative narrative should be provided, e.g. from desktop research, interviews with support workers and refugees etc.

Corruption

The paper gives some background affecting this topic, especially relating to the public support system and private support providers, which is appropriate. However, readers with some understanding of the situation would be left wondering about the effects of the EU funding provided to Turkey to deal with the refugee situation, so as to stop large numbers entering the EU. There are reports that much of the money is handed to aid organisations and the UN, not the Turkish government: ‘Ankara has always maintained the migration deal money should go directly to Turkey rather than mostly to non-governmental groups and UN agencies’ (e.g.Turkey migration deal sparks second-phase EU funding row’, https://www.ft.com/content/1e9aeb34-0100-11e8-9650-9c0ad2d7c5b5). This would make all of these them equally responsible for the all-round safekeeping and well-being of the refugees. What were the relevant terms and conditions of EU funding in this regard? More relevant details in this respect may impact on the recommendations proposed in this paper, i.e. by the EU paying more attention to the issues raised in this Report.

Author Response

The authors would like to thank reviewer #1 for the in-depth review and critique of our Brief Report on trauma affected Syrian families with children in Turkey and the impact on their educational experiences.  All the comments and recommendations for change were thoughtful and we have attempted to incorporate most if not all of the suggestions. Specifically,

Ethical issue of Children's Rights.

We agree that children are not passive entities but are rather active rights holders. A paragraph summarizing the role of the UN Convention on the Rights of the Child has been added. It highlights the general provisions to Provide, Protect and Participate. It also includes a short paragraph on Article 22 which directly speaks to the obligation of States to protect and care for refugee families with children.

Educational Interventions

We clarify that the educational programmatic components are standardized but are tailored to the make up of participants and setting.

The Study

The abstract does reference a gender analysis.  It was a qualitative analysis with this clarified in the abstract. In addition, a new sub-section was added in Section 3, briefly highlighting the methods and findings.  We believe this provides a more transition to the next section on Barriers. 

Corruption

A paragraph was added to the Conclusion discussion the EU-Turkey agreement and the direction of resources to largely administrative expenses for international civil society. We call for a review of funding and a strategy to promote collaboration.

Again we are thankful for the critique and we hope our additions and edits address the reviewers concerns.

Reviewer 2 Report

I very much enjoyed reading this Brief Report on gender related challenges in educational interventions in Turkey with Syrian refugee children. Generally the brief is in good form and my recommendations for improvement are few, but important.

The main issue is that there are a number of generalizations made, which require references. They are found in lines 63-4;68-72; 73; 92; 113-114; 119-120 ("broader community" hints at persons beyond the immediate family which the text alone does not support); 147-8; 151-4; 165-6.

Additionally, I recommend the following changes: shorten the title; adjust the font in the abstract and in lines 176-180 as it is clearly a description of A Global Voice for Autism that has been copied and pasted from elsewhere; refugees by definition are actually "in need of protection" not having fled war and violence (I recommend using and referencing a particular definition on lines 56 that is more accurate); recommend deleting the word "players" on line 87.

The final recommendation is to tailor part 4. Components of... to reflect the organization's work in Turkey already going on, rather than a broader discussion of the organization in general. It is better for the report and adds strength to the recommendations. 

Author Response

The authors would like to thank Review #2 for a very thoughtful and helpful review. We have addressed each of the items. Specifically:

References

We have added 18 additional citations to the specific items identified by the reviewer.  These indeed make the paper a much stronger manuscript. As requested, a more definitive definition of "refugee" was added with the emphasis on the importance of the need for protection.

Font

The various font discrepancies that occurred when transferring from the original manuscript to the proofs have been corrected.

Additional additions and deletions edits were also made to address the reviewers comments.

Part 4,

The items listed are presently underway and reflect the organizations efforts in the field. This was clarified as well.

Thank you again for the insightful read and critique. It is very much appreciated.

Reviewer 3 Report

This is a very interesting paper. It addresses the traumatization of children through educational interventions designed for parents. However, I have a number of concerns:

The paper's research methodology is not clearly articulated. It would be helpful if the authors could indicate how they selected the data for their study and how they arrived at their recommendations.

The paper asserts that children's learning and their sense of security are affected by trauma. But, in what way does trauma impact on a child's learning capacity and processes? Obviously, it does so in a negative way, but, presumably, if we know how trauma can affect how children's learning is changed then this might allow us to be able to devise appropriate learning strategies or to compensate in other ways to overcome their learning deficiencies.

The authors rely on the Global Voice for Autism to assist in the education of children and their parents who have been impacted by trauma. (Incidentally, those who have been forced to flee their country and have been "uprooted" have already suffered a major trauma, irrespective of any other violence perpetrated upon them.) I think there needs to be a clear distinction made between those who are suffering from autism and those who have been traumatized due to a "well-founded fear of persecution" or have been displaced due to protracted armed conflict. Autism and those who have been traumatized due to domestic abuse, in whatever form, in my opinion, should not be equated. Person's suffering from autism can, of course, also suffer from traumatizing events, but, my point is that persons who are autistic should not necessarily be combined with those who have been traumatized through any form of abuse.

The four recommendations offered seem to be focused on changes to the Turkish State's laws and policies with respect to assisting refugees, irrespective of their status, as well as calling on NGOs to build partnerships to service the needs of traumatized parents and children. But, this leaves me wondering whether there is also a role for the UN and its agencies to assist here? Perhaps the authors may wish to consider this as well.

Author Response

The authors would like to thank Reviewer #3 for a thoughtful critique of our manuscript. The follow changes have been made'

Research Methodology

We have clarified that the gender analysis was a qualitative analysis in the abstract. In addition, a new sub-section was added to Section 3, providing a brief description of the methodology as well as some preliminary results.  This was clearly needed.

Trauma, Stress and Learning

A paragraph was added to describe the physiologic changes that occur with trauma and stress on the learning process with additional citations added.

Distinction between Refugees and Refugee Children with Disabilities

The reviewer is correct that there are inherent differences clinically between refugee children with behavioral disorders and refugee children with Autism.  Though different, the program does not distinguish because trauma experiences affect them both. The educational interventions are designed to reduce stress and to improve learning trajectories and resiliency for all children they provide services to.

An additional paragraph was added to the paper to better explain the distinction and the philosophy and programmatic approach to this those children who reserve services.

Partnership with UN and NGOs

A final paragraph was added to highlight the importance of the UN Compacts on Migration and Refugees which may serve as a framework in calling for more collaborative partnerships.

Again, the authors are very grateful for the insightful review. The changes that have been made in response have made it a stronger and more compelling manuscript.

Round 2

Reviewer 3 Report

I appreciate the corrections, additions, and modifications made to this article. However, I have concerns that the material is not focused sufficiently on children per se at times. Further, the distinction between autism and traumatization is still not made to any degree. Often, my sense is that the two are conflated in the piece. In addition, the focus on one NGO, Global Voice on Autism, for data is troubling.

There are several areas of the text that need to be cleaned up as the wording is confusing.